# From Language to Goals: Inverse Reinforcement Learning for Vision-Based Instruction Following

**Justin Fu** [*] **, Anoop Korattikara, Sergey Levine, Sergio Guadarrama**
Google AI
{justinfu,kbanoop,slevine,sguada}@google.com

## Abstract

Reinforcement learning is a promising framework for solving control problems, but its use in practical situations is hampered by the fact that reward functions are often difficult to engineer. Specifying goals and tasks for autonomous machines, such as robots, is a significant challenge: conventionally, reward functions and goal states have been used to communicate objectives. But people can communicate objectives to each other simply by describing or demonstrating them. How can we build learning algorithms that will allow us to tell machines what we want them to do? In this work, we investigate the problem of grounding language commands as reward functions using inverse reinforcement learning, and argue that language-conditioned rewards are more transferable than language-conditioned policies to new environments. We propose language-conditioned reward learning (LC-RL), which grounds language commands as a reward function represented by a deep neural network. We demonstrate that our model learns rewards that transfer to novel tasks and environments on realistic, high-dimensional visual environments with natural language commands, whereas directly learning a language-conditioned policy leads to poor performance.

## 1 Introduction

While reinforcement learning provides a powerful and flexible framework for describing and solving control tasks, it requires the practitioner to specify objectives in terms of reward functions. Engineering reward functions is often done by experienced practitioners and researchers, and even then can pose a significant challenge, such as when working with complex image-based observations. While researchers have investigated alternative means of specifying objectives, such as learning from demonstration (Argall et al., 2009), or through binary preferences (Christiano et al., 2017), language is often a more natural and desirable way for humans to communicate goals.

A common approach to building natural language interfaces for reinforcement learning agents is to build language-conditioned policies

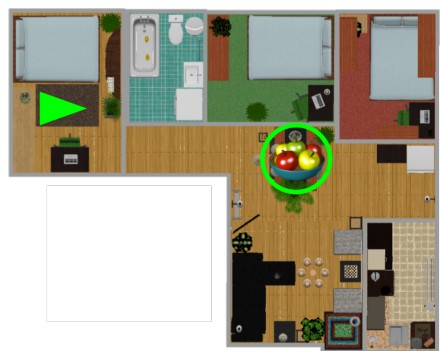

Figure 1: A task where an agent (green triangle) must execute the command "go to the fruit bowl." This is a simple example where the reward function is easier to specify than the policy.

that directly map observations and language commands to a sequence of actions that perform the desired task. However, this requires the policy to solve two challenging problems together: understanding how to plan and solve tasks in the physical world, and understanding the language command itself. The trained policy must simultaneously interpret a command and plan through

---

[*] Work done during an internship at Google AI Research

possibly complicated environment dynamics. The performance of the system then hinges entirely on its ability to generalize to new environments - if either the language interpretation or the physical control fail to generalize, the entire system will fail. We can recognize instead that the role of language in such a system is to communicate the goal, and rather than mapping language directly to policies, we propose to learn how to convert language-defined goals into reward functions. In this manner, the agent can learn how to plan and perform the task on its own via reinforcement learning, directly interacting with the environment, without relying on zero-shot transfer of policies. A simple example is shown in Figure 1, where an agent is tasked with navigating through a house. If an agent is commanded "go to the fruit bowl", a valid reward function could simply be a fruit bowl detector from first-person views of the agent. However, if we were to learn a mapping from language to actions, given the same goal description, the model would need to generate a different plan for each house.

In this work, we investigate the feasibility of grounding free-form natural language commands as reward functions using inverse reinforcement learning (IRL). Learning language-conditioned rewards poses unique computational problems. IRL methods generally require solving a reinforcement learning problem as an inner-loop (Ziebart, 2010), or rely on potentially unstable adversarial optimization procedures (Finn et al., 2016; Fu et al., 2018). This is compounded by the fact that we wish to train our model across multiple tasks, meaning the IRL problem itself is an inner-loop. In order to isolate the language-learning problem from the difficulties in solving reinforcement learning and adversarial learning problems, we base our method on an exact MaxEnt IRL (Ziebart, 2010) procedure, which requires full knowledge of environment dynamics to train a language-conditioned reward function represented by a deep neural network. While using exact IRL procedures may seem limiting, in many cases (such as indoor robotic navigation) full environment dynamics are available, and this formulation allows us to remove the difficulty of using RL from the training procedure. The crucial insight is that we can use dynamic programming methods during training to learn a reward function that maps from observations, but we do not need knowledge of dynamics to *use* the reward function, meaning during test time we can evaluate using a reinforcement learning agent without knowledge of the underlying environment dynamics. We evaluate our method on a dataset of realistic indoor house navigation and pick-and-place tasks using the SUNCG dataset, with natural language commands. We demonstrate that our approach generalizes not only to novel tasks, but also to entirely new scenes, while directly learning a language-conditioned policy leads to poor performance and fails to generalize.

## 2 RELATED WORK

A popular class of approaches to language grounding in reinforcement learning is to directly train a policy that consumes language as an input. Several works adopt a behavioral cloning approach, where the model is trained using supervised learning with language-action sequences pairs (Anderson et al., 2018; Mei et al., 2016; Sung et al., 2015). A second approach is to forego demonstrations but instead reward an agent whenever the desired task is completed (Shah et al., 2018; Misra et al., 2017; Hermann et al., 2017; Branavan et al., 2009). This approach requires reward functions (the task completion detector) to be hand-designed for the training tasks considered. Another related approach is semantic parsing, which has also been used to convert language into an executable form that corresponds to actions within an environment (Forbes et al., 2015; Misra et al., 2014; Tellex et al., 2011). In a related task to instruction following, Das et al. (2018) consider an embodied question-answering task where an agent must produce an answer to a question, where the relevant information lies within the environment. They adopt a hybrid approach, where they pretrain with supervised learning but also give the agent reward for completing intermediate tasks. Overall, our experiments show that policy-based approaches have worse generalization performance to new environments, because the policy must rely on zero-shot generalization at test time as we show in Section 6.3. While in this paper we argue for the performance benefits of a reward-based approach, a reason one may want to adopt a policy-based approach over a reward-based one is if one cannot run RL to train a new policy in a new environment, such as for time or safety reasons.

A second approach to the language grounding problem is to learn a mapping from language to reward functions. There are several other works that apply IRL or IRL-like procedures to the problem of language grounding. Perhaps most closely related to our work is MacGlashan et al. (2015), which also aims to learn a language-conditioned reward function via IRL. However, this method

requires an extensively hand-designed, symbolic reward function class, whereas we use generic, differentiable function approximators that can handle arbitrary observations, including raw images. Bahdanau et al. (2018); Tung et al. (2018) also learn language-conditioned reward functions, but do not perform IRL, meaning that the objective does not correspond to matching the expert's trajectory distribution. Tung et al. (2018) train a task-completion classifier, but do not evaluate their reward on control problems. The strategy they use is similar to directly regressing onto a ground-truth reward function, which we include a comparison to in Section 6 to as an oracle baseline. Bahdanau et al. (2018) adopt an adversarial approach similar to GAIL (Ho & Ermon, 2016), and use the learned discriminator as the reward function. While this produces a reward function, it does not provide any guarantees that the resulting reward function can be reoptimized in new environments to yield behavior similar to the expert. We believe our work is the first to apply language-conditioned inverse reinforcement learning to environments with image observations and deep neural networks, and we show that our rewards generalize to novel tasks and environments.

## 3  BACKGROUND

We build off of the MaxEnt IRL model (Ziebart et al., 2008), which considers an entropy-regularized Markov decision process (MDP), defined by the tuple $(\mathcal{S}, \mathcal{A}, \mathcal{T}, r, \gamma, \rho_0)$. $\mathcal{S}, \mathcal{A}$ are the state and action spaces respectively and $\gamma \in (0, 1)$ is the discount factor. $\mathcal{T}(s'|s, a)$ represents the transition distribution or dynamics. We additionally consider partially-observed environments, where each state is associated with an observation within an observations space $o \in \mathcal{O}$.

The goal of "forward" reinforcement learning is to find the optimal policy $\pi^*$. Let $r(\tau) = \sum_{t=0}^{T} \gamma^t r(s_t, a_t)$ denote the returns of a trajectory, where $\tau$ denotes a sequence of states and actions $(s_0, a_0, ...s_T, a_T)$. The MaxEnt RL objective is then to find $\pi^* = \arg\max_\pi E_{\tau\sim\pi}[r(\tau) + H(\tau)]$.

Inverse reinforcement learning (IRL) seeks to infer the reward function $r(s, a)$ given a set of expert demonstrations $\mathcal{D} = \{\tau_1, ..., \tau_N\}$,. In IRL, we assume the demonstrations are drawn from an optimal policy $\pi^*(a|s)$. We can interpret the IRL problem as solving the maximum likelihood problem:

$$\max_\theta E_{\tau\sim\mathcal{D}} \left[\log p_\theta(\tau)\right] \ , \tag{1}$$

In the MaxEnt IRL framework, optimal trajectories are observed with probabilities proportional to the exponentiated returns, meaning $p(\tau) \propto \exp\{r(\tau)\}$ (Ziebart et al., 2008). Thus, learning a reward function $r_\theta(\tau)$ is equivalent to fitting an energy-based model $p_\theta(\tau) \propto \exp\{r_\theta(\tau)\}$ to the maximum likelihood objective in Eqn 1. The gradient to update the reward function is (Ziebart, 2010):

$$\nabla_\theta E[\log p_\theta(\tau)] = \sum_{s,a}(\rho^\mathcal{D}(s, a) - \rho_\theta^*(s, a))\nabla_\theta r_\theta(s, a) \ , \tag{2}$$

where $\rho^\mathcal{D}(s, a)$ represents the state-action marginal of the demonstrations, and $\rho_\theta^*(s, a)$ represents the state-action marginal of the optimal policy under reward $r_\theta(s, a)$.

## 4  MULTI-TASK IRL

A unique challenge of the language-conditioned IRL problem, compared to standard IRL, is that the goal is to learn a reward function that generalizes across multiple tasks. While standard IRL methods are typically trained and evaluated on the same task, we want our language-conditioned reward function to produce correct behavior when presented with new tasks. Several previous works consider a multi-task scenario, such as in a Bayesian or meta-learning setting (Li & Burdick, 2017; Dimitrakakis & Rothkopf, 2012; Choi & Kim, 2012). We adopt a similar approach adapted for the language-IRL problem, and formalize the notion of a task, denoted by $\xi$, as an MDP, where individual tasks may not share the same state spaces, dynamics or reward functions. Each task is associated with a context $c_\xi$ which is a unique identifier (i.e. an indicator vector) for that task. Thus, we wish to optimize the following multi-task objective, where $\tau_\xi$ denotes expert demonstrations for that task:

$$\max_\theta E_\xi[E_{\tau_\xi}[\log p_\theta(\tau_\xi, c_\xi)]] \tag{3}$$

---

**Algorithm 1** Language-Conditioned Reward Learning (LC-RL)

---
1: Obtain expert demonstrations and language describing the goal.
2: Initialize reward function $r_\theta$.
3: **for** step $t$ in $\{1, \ldots, N\}$ **do**
4:      Sample task $\xi$, demonstrations $d_\xi$ , and language $\mathcal{L}_\xi$.
5:      Compute optimal $q^*(s, a)$ using q-iteration and $\rho^*(s, a)$ using the forward algorithm.
6:      Update reward $r_\theta$ with the gradient $(\rho^{d_\xi}(s, a) - \rho^*(s, a))\nabla r_\theta(o, a, \mathcal{L}_\xi)$
7: **end for**

---

In order to optimize this objective, we first require that all tasks share the same observation space and action space, and the reward to be a function of the observation, rather than of the state. For example, in our experiments, all observations are in the form of 32x24 images taken from simulated houses, but the state space for each house is allowed to differ (i.e., the houses have different layouts). This means the same reward can be used across all MDPs even though the state spaces differ.

Second, we share the reward function across all tasks, but substitute a language command $\mathcal{L}_\xi$ as a proxy for the context $c_\xi$, resulting in a model $p_\theta(\tau_\xi, \mathcal{L}_\xi)$ that takes as input language, states, and actions. For computational efficiency we run stochastic gradient descent on the objective in Eqn. 3 by sampling over the set of environments on each iteration.

## 5 LANGUAGE-CONDITIONED REWARD LEARNING (LC-RL)

We learn language-conditioned reward functions using maximum causal entropy IRL, adapted for a multi-task setting and rewards represented by language-conditioned convolutional neural networks. While during training we use dynamic programming methods that require dynamics knowledge, we do not need knowledge of dynamics to evaluate the reward function. Thus, at test time we can use standard model-free RL algorithms to learn the task from the inferred reward function in new environments. Our algorithm is briefly summarized in Algorithm 1.

### 5.1 COMPUTING EXACT IRL GRADIENT UPDATES

In order to take gradient steps on the objective of Eqn. 3, we update our reward function in terms of the Maximum Entropy IRL gradient (Ziebart, 2010) according to Eqn. 2. The stochastic gradient update (for a single task $\xi$) adapted to our case is:

$$\nabla_\theta E_\tau[\log p_\theta(\tau_\xi, \mathcal{L}_\xi)] = \sum_{s,a}(\rho^d(s, a) - \rho^*_\theta(s, a))\nabla_\theta r_\theta(o(s), a, \mathcal{L}_\xi)$$

Where $o(s)$ denotes the observation for state $s$. Note that during training we need access to the ground truth states $s$. While the update depends on the underlying state, the reward itself is only a function of the observation, the action, and the language. This enables us to evaluate the reward without knowing the underlying state space and dynamics of the environment. While requiring dynamics knowledge during training may seem limiting, in practice many environments we may wish to train a robot in can easily be mapped. This training strategy is analogous to training a robot in only known environments such as a laboratory, but the resulting reward can be used in unknown environments.

In order to compute $\rho^*_\theta(s, a)$, one normally has to first compute the optimal policy with respect to reward $r_\theta(o, a)$ using reinforcement learning, and then compute the occupancy measure using the forward algorithm for Markov chains to compute the state visitation distributions at each time-step. Because this embeds a difficult RL optimization problem within nested inner-loops, this quickly becomes computationally intractable. Thus, we train in tabular environments with known dynamics, where we can compute optimal policies exactly using Q-iteration. However, we emphasize that this is only a training time restriction, and knowledge of dynamics is not required to evaluate the rewards.

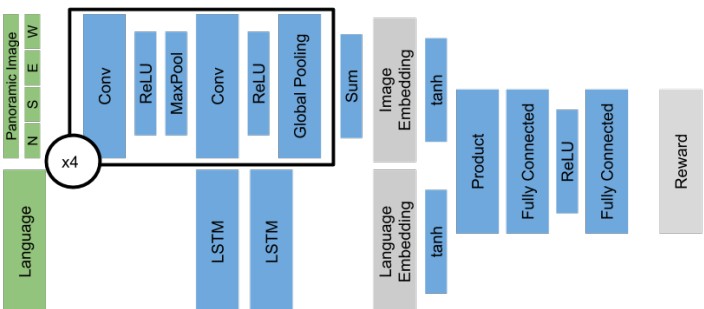

Figure 2: Our reward function architecture. Our network receives as input a panoramic semantic image (4 views) and a language command represented as a sequence of one-hot word vectors, and outputs a scalar reward.

## 5.2 ARCHITECTURE

Our network architecture is shown in Figure 2. The network has two main modalities of input: a variable-length language input represented by a sequence of one-hot vectors (one vector for each tokenized word), and a panoramic image observation.

The language embedding is formed by processing the language input sequence through an LSTM network, and the final time-step of the topmost layer is used as a fixed-dimensional embedding $e_{\text{language}}$ of the input command.

The agent receives image observations in the form of four 32x24 image observations, one for each cardinal direction view (N, S, E, W). The convolutional neural network (CNN) consists of a sequence of convolutional and max pool layers, with the final operation being a channel-wise global pooling operation that produces an image embedding of the same length as the language embedding. Each image is passed through an identical CNN with shared weights, and the outputs are summed together to form the image embedding. That is, $e_{\text{image}} = \text{CNN}(\text{img}_N) + \text{CNN}(\text{img}_S) + \text{CNN}(\text{img}_E) + \text{CNN}(\text{img}_W)$.

Finally, these two embeddings are element-wise multiplied and passed through a fully-connected network (FC) to produce a reward output. Letting $\odot$ denote elementwise multiplication, we have $r = \text{FC}(e_{\text{image}} \odot e_{\text{language}})$.

We found that the max global-pooling architecture in the CNN was able to select out objects from a scene and allow the language embedding to modulate which features to attend to. We selected our architecture via a hyper-parameter search, and found that the choice of using an element-wise multiplication versus a concatenation for combining embeddings had no appreciable performance difference, but a global pooling architecture performed significantly better than using a fully connected layer at the end of the CNN.

## 6 EVALUATION

We evaluate our method within a collection of simulated indoor house environments, built on top of the SUNCG (Song et al., 2017) dataset. The SUNCG dataset provides a large repository of complex and realistic 3D environments which we find very suitable for our goals. An example task from our environment is shown in Figures 3 and 4. An example of a successful execution of a task is showin in Fig. 5.

## 6.1 ENVIRONMENT

We consider two typical kinds of tasks that an indoor robot may wish to perform:

- **Navigation (NAV)**: In the navigation task, the agent is given a location which corresponds to a room or object, and the agent must navigate through the house to reach the target location. For example, in Fig. 3, the target could be "cup" or "laptop" or "living room".

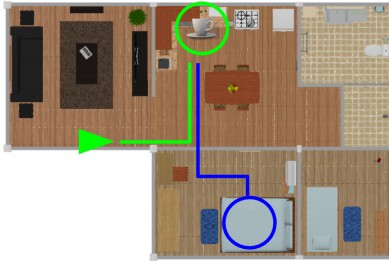

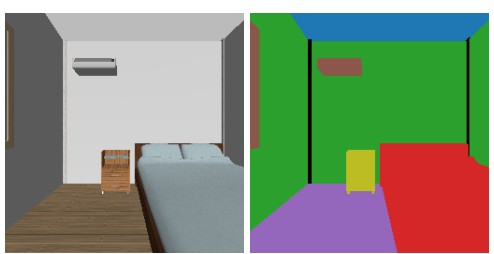

Figure 3: An example task. The green segment corresponds to the solution of a NAV task, "go to the cup", where the cup is circled in green. The green plus blue segments represents a path for the PICK task, "move the cup to the bed", where the bed is circled in blue.

Figure 4: Example first-person RGB (left) and semantic (right) images from the bedroom inside the house depicted in Figure 3. We only use the semantic labels as input to our model.

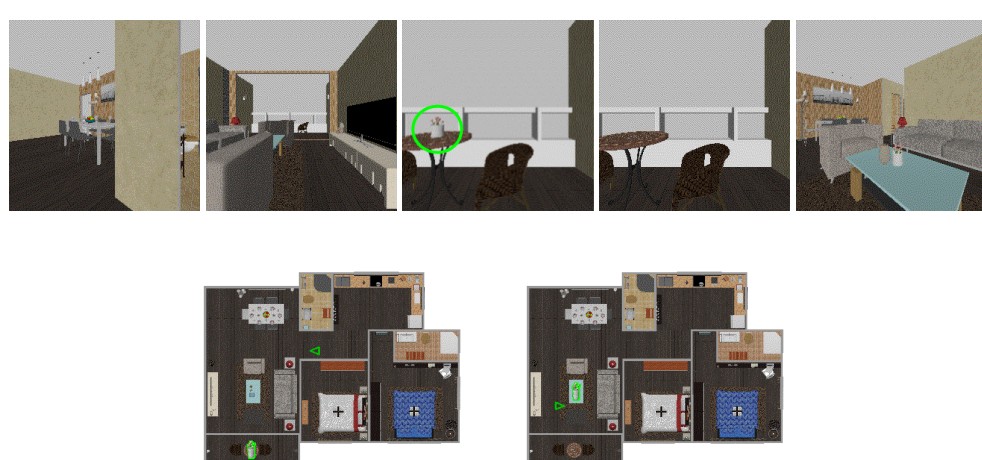

Figure 5: *Top row*: First-person view of an agent executing the task: "move vase to living room". The vase is circled in green in the 3rd image. *Bottom row*: A bird's eye view of the initial (left) and final (right) positions of the agent (green triangle) and the vase (green outline).

- **Pick-and-place (PICK)**: In the pick and place task, the agent must move an object from one location to another. For example, in Fig. 1 the task could be to move the cup from to the sink to the kitchen table.

Each environment corresponds to one 3D scene, which is discretized into a grid to form a tabular environment where the grid coordinates plus agent orientation (N, S, E, W) correspond to the state of the agent. The agent receives observations with two components: one is a free-form language command, and one is a first-person panoramic image of the environment. Because the agent can move objects without directly looking at them, the panoramic view gives the agent a full view of its surroundings. The panoramic image is formed from 4 semantic image observations, one for each orientation of the agent. Each semantic image observation is 32x32 pixels and contains 61 channels, one per semantic image class. Each agent is equipped with 4 actions: step forward one grid tile, turn left or right, or interact with an object.

We generate language commands based on a preset grammar, and using names of objects and locations associated with the task. These are of the form "go to X" for NAV tasks, or "move X to Y" for PICK tasks, where X and Y stand for names of locations and objects within the environment. We explicitly do not use step-by-step instruction language such as "turn left, walk down the hallway, go through the door", as these commands remove the planning aspect of the problem and tell the agent directly which actions to take in order to solve the problem.

The interact action only has meaning within the PICK task. Executing this action will either pick up an object if the agent is within a 1 meter of an object, or drop an object if the agent is currently

holding an object. To limit the size of the state space, within a single task, there is only one object an agent may interact with and two locations the object can be in. This setup only increases the size of the state-space by a factor of 3. However, different objects may be placed in different locations across environments, meaning the model still must learn to detect the object in the correct location rather than memorizing the specific object and location associated with a single task.

## 6.2 EXPERIMENTAL PROCEDURE

In order to evaluate how well different methods generalize, we split our dataset of tasks into three segments: a training set, and two test sets – "task" and "house". The "task" test set contains tasks within the same houses as training, but requires the agent to interact with novel combinations of objects and locations. The "house" test set contains tasks on entirely new houses that were not in the training set. The purpose of this split is to investigate varying degrees of generalization: the "task" test set requires the model to execute novel language commands, but using landmarks and objects which were seen during training. The "house" test set adds another layer of difficulty, requiring the model to detect familiar objects situated in entirely new scenes.

In total, our dataset contains 1413 tasks (716 PICK, 697 NAV). Across all tasks, there are 14 objects, and 76 different house layouts. There are 1004 tasks (71%) in the training set, 236 (17%) in the "task" test set, and 173 (12%) in the "house" test set.

We evaluate two methods for reward learning. LC-RL refers to the language-conditioned IRL method outlined in Section 5, which takes as input demonstration and language pairs and learns a shared reward function across all tasks. In particular, we use 10 demonstrations per task, sampled from the computed optimal policy. To provide an upper bound for performance, we can also regress directly onto the ground-truth rewards, a method we label as "Reward Regression". While this is not possible in a practical scenario, this evaluation serves to show oracle performance on our task. Note that our method does *not* require access to ground-truth rewards, and only uses demonstrations.

Success rates for each reward-learning method are evaluated using two policy learning procedures. Q-iteration (QI) computes the optimal policy exactly using dynamic programming, which we report in Table 1. We also experiment with reoptimizing the learned reward using DQN (Mnih et al., 2015), a sample-based RL method that does not require ground-truth knowledge of the environment dynamics. We use the position, orientation, and whether an object is held as the observation (this is identical to the state representation). This experiment represents the testing use-case where we can evaluate the reward at test-time in novel, unmapped environments despite the fact that at training time we require dynamics knowledge. However, because the probability that the random policy receives rewards on our task is tiny, we found that epsilon-greedy exploration was not enough. Thus, we also report results using a reward shaping term with a state-based potential equal to the optimal value function (Ng et al., 1999). We note that this shaping term does require dynamics knowledge compute, but we include this result to highlight the difficulty of the RL problem even if reward learning is done properly.

We also compare against two baselines derived from GAIL (Ho & Ermon, 2016), using the learned discriminator as the "reward" function. We first compare to AGILE (Bahdanau et al., 2018), which modifies GAIL to use a goal-based discriminator and false negative filtering, using DQN as a policy optimizer and $\rho = 0.25$. We found it difficult to learn rewards using a reinforcement learning-based policy optimizer, and the model was only able to solve the simpler NAV environments. This experiment emphasizes the gap between using a sampling-based solver and an exact solver during the training of reward-based methods. To create a more fair comparison, we also compare against GAIL using a dynamic programming solver (labeled GAIL-Exact), and we see that the performance is comparable to LC-RL on training environments, but performs significantly worse on test environments. These results are in line with our intuitions - GAIL and IRL are equivalent in training scenarios (Ho & Ermon, 2016), but the discriminator of GAIL does not correspond to the true reward function, and thus performs worse when evaluated in novel environments.

In order compare against a policy-learning approach, we compare against an optimal behavioral cloning baseline. We train the optimal cloning baseline by computing the exact optimal policy using Q-iteration, and perform supervised learning to regress directly on to the optimal action probabilities. To make a fair comparison, we keep the policy architecture identical to the reward architecture, except we add two additional inputs: the orientation indicator and and indicator on whether the ob-

Table 1: Success rates (in percentages) across task categories. Each result is averaged over 3 seeds. Test-Task refers to testing on novel tasks within the same houses as training, whereas Test-House refers to testing novel tasks in novel houses. The AGILE method is described in (Bahdanau et al., 2018)

| | Train | | | Test-Task | | | Test-House | | |
|---|---|---|---|---|---|---|---|---|---|
| | PICK | NAV | Total | PICK | NAV | Total | PICK | NAV | Total |
| Optimal Policy Cloning | 20.7 | 61.6 | 40.3 | 10.1 | 29.4 | 19.6 | 0.0 | 17.2 | 8.5 |
| AGILE | 0.0 | 40.9 | 18.0 | 0.0 | 34.1 | 16.8 | 0.0 | 30.6 | 15.1 |
| GAIL-Exact | 59.4 | **73.5** | **66.9** | 49.1 | **50.4** | 49.8 | 23.5 | 35.4 | 28.3 |
| LC-RL (ours) | **63.8** | 69.7 | **66.9** | **56.7** | 47.8 | **51.9** | **32.1** | **39.4** | **36.4** |
| Reward Reg. (Oracle) | 87.0 | 85.0 | 86.1 | 82.5 | 67.0 | 74.1 | 70.6 | 62.3 | 65.7 |

Table 2: Success rates (in percentages) on using DQN to re-optimize learned rewards. For reference, we also include Q-iteration results (labeled QI) from Table 1 as an oracle comparison.

| | Shaping | Train | Test-Task | Test-House |
|---|---|---|---|---|
| LC-RL (DQN) | Yes | 12.9 | 14.1 | 14.9 |
| | No | 8.0 | 7.7 | 8.0 |
| LC-RL (QI) | - | 66.9 | 51.9 | 36.4 |
| Reward Regression (DQN) | Yes | 54.7 | 58.9 | 57.5 |
| | No | 7.5 | 8.3 | 9.2 |
| Reward Regression (QI) | - | 86.1 | 74.1 | 65.7 |

ject (during PICK tasks) is held by the agent or not. Each indicator is transformed by an embedding lookup, and all embeddings are element-wise multiplied along with with the language and image embeddings in the original architecture.

## 6.3 EXPERIMENTAL RESULTS

Our main experimental results on reward learning are reported in Table 1, and experiments in re-optimizing the learned reward function are reported in Table 2. Qualitative results with diagrams of learned reward functions can be found in Appendix B. Additional supplementary material can be viewed at `https://sites.google.com/view/language-irl`, and experiment hyperparameters are detailed in Appendix A.

We found that both LC-RL and Reward Regression were able to learn reward functions which generalize to both novel tasks and novel house layouts, and both achieve significant performance over the policy-based approach. As expected, we found that Reward Regression has superior performance when compared to LC-RL, due to the fact that it uses oracle ground-truth supervision.

We include examples of learned reward functions for both methods in Appendix B. We found that a common error made by the learned rewards, aside from simply misidentifying objects and locations, was rewarding the agent for reaching the goal position without placing the object down on PICK tasks. This is reflected in the results as the performance on PICK tasks is much lower than that of NAV tasks. Additionally, there is some ambiguity in the language commands, as the same environment may contain multiple copies of a single object or location, and we do not consider the case when agents can ask for additional clarification (for example, there are 2 beds in Fig. 3).

We observed especially poor performance from the cloning baseline on both training as well as testing environments, even though it was trained by directly regressing onto the optimal policy. We suspect that it is significantly more difficult for the cloning baseline to learn across multiple environments. Our language is high-level and only consists of descriptions of the goal task (such as "move the cup to the bathroom") rather than step-by-step instructions used in other work such as Mei et al. (2016) that allow the policy to follow a sequence of instructions. This makes the task

much more difficult for a policy-learning agent as it needs to learn a mapping from language to house layouts instead of blindly following the actions specified in the language command.

Regarding re-optimization of the learned rewards, we found that DQN with epsilon-greedy exploration alone achieved poor performance compared to the exact solver and comparable performance to the cloning baseline (however, note that our cloning baseline was regressing onto the exact optimal actions). Adding a shaping term based on the value-function improves results, but computing this shaping term requires ground-truth knowledge of the environment dynamics. We also note that it appears that rewards learned through regression are easier to re-optimize than rewards learned through IRL. One explanation for this is that IRL rewards appear more "noisy" (for example, see reward plots in Appendix B, because small variations in the reward may not affect the trajectories taken by the optimal policy if a large reward occurs at the goal position. However, while RL is training it may never see the large reward and thus is heavily influenced by small, spurious variations in the reward. Nevertheless, with proper exploration methods, we believe that language-conditioned reward learning provides a performant and conceptually simple method for grounding language as concrete tasks an agent can perform within an interactive environment.

## 7    CONCLUSION

In this paper, we introduced LC-RL, an algorithm for scalable training of language-conditioned reward functions represented by neural networks. Our method restricts training to tractable domains with known dynamics, but learns a reward function which can be used with standard RL methods in environments with unknown dynamics. We demonstrate that the reward-learning approach to instruction following outperforms the policy-learning when evaluated in test environments, because the reward-learning enables an agent to learn and interact within the test environment rather than relying on zero-shot policy transfer.

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

APPENDICES

## A  EXPERIMENT HYPERPARAMETERS

For our environment, we give the agent a time limit 30 time-steps to complete a task. For the purpose of reward regression and generating demonstrations, the environment gives a reward of 10 when the agent successfully completes the task. We sample demonstrations from the optimal policy using this ground-truth reward. Our MDP solvers use a discount of $\gamma = 0.99$.

For our model, we used 10 demonstrations per environment to train IRL, and optimized with Adam using a learning rate of $5 * 10^{-4}$.

For our convolutional neural network, we used a 5x5 convolution with 16 filters, followed by a 3x3 convolution with 32 filters. The size of each embedding was 32. The final fully connected layer had sizes of 32, and 1 (for the final output). We did not find significant performance differences increasing the number of filters or embedding sizes.

We selected our architecture through a hyper-parameter sweep, with train and test accuracies presented below (averaged over 3 seeds each). The main architectures we swept through were whether to produce the image embedding via a global pooling layer (labeled Pooling) versus a single fully connected layer (labeled FC) versus FiLM (Perez et al., 2018), and whether to combine the language and image embeddings using a point-wise multiplication or a pooling operation (labeled Mult) versus concatenation (labeled Concat).

|  | Train | Test-Task |
|---|---|---|
| Pooling, Mult | 67.3 | 49.7 |
| Pooling, Add | 64.1 | 45.5 |
| Pooling, Concat | 65.8 | 48.5 |
| FiLM, Mult | 60.6 | 43.2 |
| FiLM, Add | 67.0 | 49.1 |
| FiLM, Concat | 63.2 | 44.6 |
| FC, Mult | 53.1 | 38.4 |
| FC, Add | 55.2 | 39.1 |
| FC, Concat | 48.1 | 35.2 |

We also conducted a single ablation study over the size of the image below, using the Pooling, Mult architecture. We found that the impact of the image size between (32 by 24) and (64 by 64) was negligible, so we selected the smaller image size for computational reasons.

|  | Train | Test-Task |
|---|---|---|
| 32 by 24 | 67.3 | 49.7 |
| 64 by 64 | 70.6 | 51.1 |

## B  LEARNED REWARDS AND QUALITATIVE RESULTS

Below is an example of a learned reward (and the computed value function) from our IRL model. The task presented here is to bring the fruit bowl (green arrow) to the bathroom (red arrow). In the top row, we plot the reward function, and in the bottom row we plot the resulting value function. The left column shows the rewards/values before the object (fruit bowl) is acquired, and the right column shows the rewards/values after. Note that before the object is acquired, the value directs the agent to the fruit bowl, and once the object is found, the value directs the agent to the bathroom.

In these figures, a blue shaded square means a high values and red means low. The green-outlined tiles correspond to all locations within a 1-meter radius of the fruit bowl, or all tiles an agent can pick up the bowl from. The red-outlined tile likewise represents all tiles within a 1-meter radius of the drop-off location in the bathroom. The blue-outlined square represents the starting location of the agent.

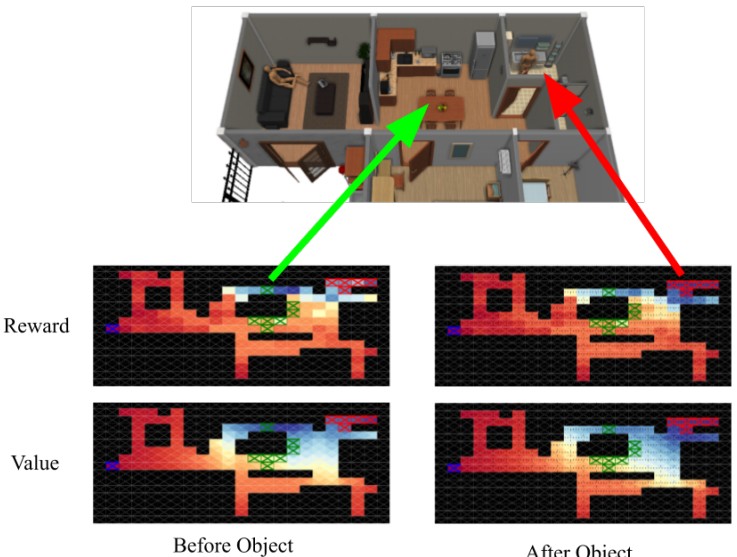

Next, we show rewards learned by 3 different methods: inverse reinforcement learning (IRL), GAIL, and reward regression. Again, low rewards are denoted by red squares and high rewards are denoted by blue squares. For each task, we also include a birds-eye view of task, where the object is highlighted in green and the agent is denoted by a green triangle.

In general, we find that rewards learned by IRL and GAIL tend to be noisy and contain small artifacts. This is not unexpected, as both of these methods are sample-based and observe demonstrations instead of ground-truth rewards as supervision. We believe that such artifacts are detrimental when using RL to reoptimize the learned reward, as without adequate exploration RL cannot find the large reward at the true goal state, and instead ends up finding local minima.

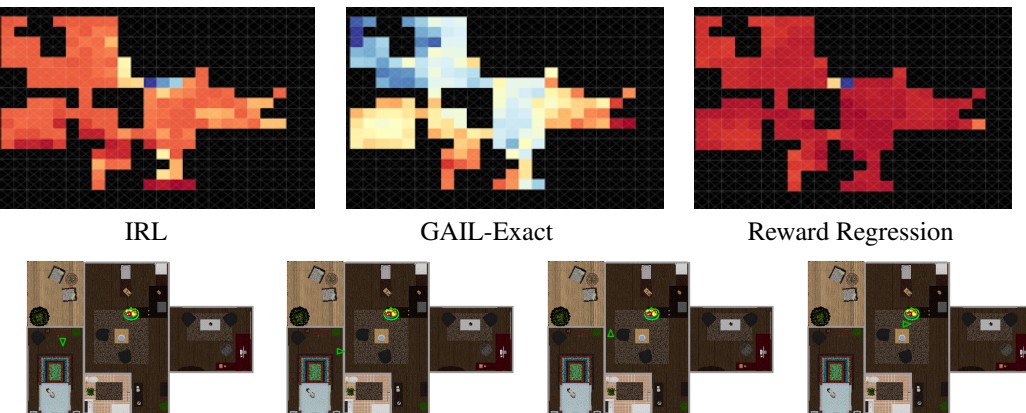

Figure 6: Learned rewards and a corresponding birds-eye view rollout for the task "go to fruit bowl".

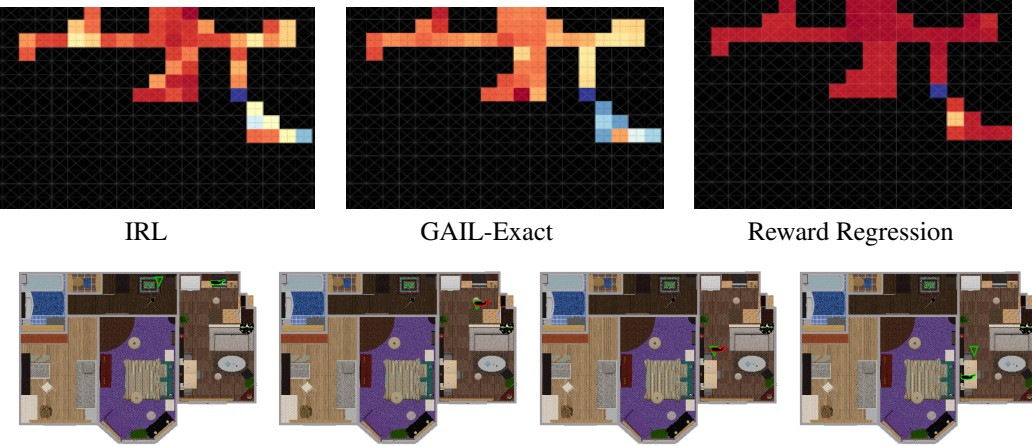

IRL             GAIL-Exact             Reward Regression

Figure 7: Learned rewards and a corresponding birds-eye view rollout for the task "move pan to living room".

## C  OBSERVATION CACHING

The running time for dynamic programming algorithms for solving MDPs (such as q-iteration) scales with the size of the state space, and in our environments we found reward evaluation becoming a major bottleneck in runtime. One major optimization we make to our algorithm is to cache computation on repeated observations. Computation-wise, the main bottleneck in Algorithm 1 is evaluating and back-propagating the reward function at all states and actions, rather than Q-iteration itself (even though it carries a cubic run-time dependency on the size of the state space). However, in many environments this is extremely wasteful. For example, in the house depicted in Figure 3, information about where the cup is located must be included in the state. However, if our observations are images of what the robot sees, whether the cup is in the kitchen or in the bathroom has no impact on the images inside the living room. This means that we should only need to evaluate our reward in each living room images once for both locations of the cup. A major factor in speeding up our computation was to cache such repeated computation. In practice we found this to be significant, resulting in 10-100 times speedups on reward computation depending on the structure of the environment.

