# OpenReview forum: "From Language to Goals: Inverse Reinforcement Learning for Vision-Based Instruction Following"
_ICLR.cc/2019/Conference_

### Official Review · AnonReviewer2 · 2018-11-02
**Comparison to Imitation Learning (not just naive BC)!**

**Rating:** 9
**Confidence:** 5

**Review:**

Summary:

This paper proposes learning reward functions via inverse reinforcement learning (IRL) for vision-based instruction following tasks like "go to the cup". The agent receives the language instruction (generated via grammar templates) and a set of four images (corresponding to four cardinal directions) from virtual cameras mounted on the agent as input at every time step and its aim is either to 1. navigate to the goal location (navigation task) or 2. move an object from one place to another (pick task).

The really interesting part in this paper is learning reward functions such that they generalize across different tasks and environments (e.g. indoor home layouts). This differentiates it from the standard IRL setting where reward functions are learnt and then policies optimized on this reward function on the *same* environment.

In order to generalize across tasks and environments a slight modification to the max-ent IRL gradient equations are made: 1. Similar to meta-learning the gradient is taken with respect to multiple tasks (in a sampling-based manner) and 2. Making the reward function a function of not just states and actions but also language context. The overall algorithm (Algorithm 1) is simple and the critical step of computing an optimal policy to compute the IRL gradient is done by assuming that one has access to full state and dynamics and essentially running a planner on the MDP. This assumption is not unreasonable since in a simulator one has access to the full dynamics and can hence one can compute the optimal trajectories by planning.

Experiments are presented on the SUNCG dataset of indoor environments. Two baselines are presented: One using behavior cloning (BC) and an oracle baseline which simply regresses to the ground truth reward function which is expected to be an upper bound of performance. Then DQN is used (with and without reward shaping) using the learnt reward functions to learn policies which are shown to have better performance on different tasks.

Comments and Questions:

- The paper is generally well-written and easy to understand. Thanks!

- The idea of using IRL to learn generalizable reward functions to learn policies so as to aid transfer between environments in such vision-language navigation tasks is interesting and clearly shows benefits to behavior cloning.

- One of my main concerns (and an interesting question that this paper naturally raises) is how does this approach compare to imitation learning (not vanilla behavior cloning which is straight-up supervised learning and has been theoretically and empirically shown to have worse performance due to distribution shifts. See Ross and Bagnell, 2011, Ross, Gordon, Bagnell 2012 (DAgger, Ross and Bagnell 2014 (AggreVate), Chang et al., 2015 (LOLS), etc). If the same budget of 10 demonstrations per environment is used via DAgger (where say each iteration of DAgger gets say 2 or 3 demonstrations until the budget is exhausted) how does it compare? Note online version of DAgger has already been used in similar settings in "Vision-and-Language Navigation: Interpreting visually-grounded navigation instructions in real environments" by Anderson et al, CVPR 2018. The main difference from their setting is that this paper considers higher level tasks instead of taking as input low-level turn-by-turn language inputs.

- The following papers are relevant and should be cited and discussed:
"Vision-and-Language Navigation: Interpreting visually-grounded navigation instructions in real environments" by Anderson et al, CVPR 2018.

"Embodied Question Answering", Das et al, CVPR 2018.

Update:
------------
After looking at other reviews and author rebuttals to all reviews I am raising my grade.

---

> ### Author Response · Authors · 2018-11-18
> **Thank you for your constructive comments**
>
> Thank you for your comments and pointing out additional related work. We agree that the main contribution of the paper was to demonstrate the effectiveness of reward learning and its ability to transfer better across visually complex environments than other imitation learning approaches. We have also included a baseline against DAgger and additional baselines against GAIL-based approaches in table 1, section 6.2.
>
> “One of my main concerns …. is how does this approach compare to imitation learning”
> Our original behavioral cloning baseline (table 1, section 6.2) runs supervised learning to match the optimal policy (computed via planning/DP) at all states, and not from demonstrations. We believe this provides an upper bound on the performance of any policy-based method, as it is receiving expert advice at every state (in contrast with DAgger-based approaches which only receive this at queried states), and thus distribution shift is not a large concern. Perhaps the name "behavioral cloning" - we have relabeled this baseline as “optimal policy cloning”.
>
> We have updated the related work section to include additional discussion of the works the reviewer has mentioned mentioned. Anderson et. al , as mentioned, concentrates on lower level tasks and policy-based learning. We believe our policy-learning baseline is the strongest baseline we could make for that class of methods and thus provides an accurate comparison, since imitation learning methods do not typically assume access to optimal actions at each state. This is unrealistic, but we believe it provides an upper bound on the performance of any direct imitation method. Das et. al. is also certainly related although they consider a different task based on question-answering. Both of these methods also adopt policy-learning approaches, rather than learning rewards.

---

> > ### Comment · AnonReviewer2 · 2018-11-21
> > **Thanks for the BC clarification!**
> >
> > If BC is being done from *all* states then yes I completely agree that you won't have the usual issues. It is indeed then a very strong baseline.

---

### Official Review · AnonReviewer3 · 2018-11-02
**Interesting paper, but results are not very convincing.**

**Rating:** 5
**Confidence:** 4

**Review:**

Paper Summary: This paper studies the inverse reinforcement learning problem for language-based navigation. Given the panorama image as its observation, language embedding as its goal, a deep neural network architecture is proposed to predict the reward function from the input observation and goal. Maximum causal entropy IRL has been adopted to learn such language-conditioned reward function. This paper used the SUNCG environment for experiments and designed two tasks (navigation and pick-and-place) for evaluation.

==
Novelty & Significance:
This paper studies a very interesting topic in reinforcement learning and the problem has potential usage when training robot agent in the real world.

==
Quality:
Overall, reviewer feels that the experimental results are not very strong. Some of the points are not clearly presented.

Firstly, is not very clear whether the argument made in the abstract “directly learning a language-conditioned policy leads to a poor performance” is justified or not. Please clarify this point in the rebuttal.

Secondly, Table 1 and Table 2 can only be treated as ablation studies. The “reward-regression” is not a baseline but more about a oracle model.
Is it possible to compare against some recent work such as Tung et al 2018 or Bahdanau et al 2018? Otherwise, it is not very clear whether the proposed approach is the state-of-the-art or not.

Thirdly, using the panorama image as observation seems not a practical choice. Is it possible to provide some ablation studies or discussions on the performance over number of views?

Finally, the architecture design is not well-justified. Why not using pre-trained image classifiers (or DQN encoder) as feature extractor (or finetune the model from pre-trained image classifier)? The actual resolution (32 x 24 x 3) in the paper looks a bit unusual.

One more thing, the url provided in the paper directs to an empty project page.

If these concerns can be addressed in the rebuttal, reviewer is happy to re-evaluate (e.g., raise scores) this work.

---

> ### Author Response · Authors · 2018-11-18
> **We have included additional experiments as requested**
>
> Thank you for the detailed review. We have run additional experiments as requested, and respond to the concerns raised below.
>
> “Firstly, is not very clear whether the argument made in the abstract  “directly learning a language-conditioned policy leads to a poor performance” is justified or not “
> This claim is supported by our experiments against the behavioral cloning baseline. We have also added an additional comparison against DAgger (table 1, section 6.2), which is the strategy used by several previous works. For this baseline, we regress directly onto the optimal actions at every single state (which we compute using an oracle), and thus we believe this is an upper-bound on performance for a policy-based method. This baseline is stronger than supervised learning from language and demonstration pairs, as used by papers mentioned in the related work, and for methods such as DAgger and active learning, because every single state in the environment is part of the training set.
>
> “Is it possible to compare against some recent work such as Tung et al 2018 or Bahdanau et al 2018? Otherwise, it is not very clear whether the proposed approach is the state-of-the-art or not.”
> We have included a baseline against Bahdanau et. al. 2018 and a variant which uses an exact policy solver, as the original Bahdanau paper uses an RL solver as an inner-loop for policy optimization, rather than planning/dynamic programming. The results are shown in Table 1 in Section 6.2. Our method achieves substantially better performance compared to Bahdanau et. al 2018 in this comparison.
>
> “Thirdly, using the panorama image as observation seems not a practical choice. Is it possible to provide some ablation studies or discussions on the performance over number of views? “
> We have updated section 6.1 to include discussion of the choice of panoramic images.  In our experiments, we did notice a significant degradation in performance without the panorama observations, because the agent can sometimes pick/place up objects without seeing the goal location, which confuses the reward regressor. In practice, however, a panorama image can be obtained on a robot by having multiple cameras mounted on a robot (which many robots already have). Due to time constraints, we did not have a chance to run this ablation study, but we will include it in the final version.
>
> “Finally, the architecture design is not well-justified. Why not using pre-trained image classifiers (or DQN encoder) as feature extractor (or finetune the model from pre-trained image classifier)?”
> We have updated appendix A to show an ablation study over architectures. We did not find that changing the architecture led to significant performance differences, and thus we believe the architecture is not the main bottleneck in performance. We decided to pick a simple architecture which at the same time performed well.
>
> “The actual resolution (32 x 24 x 3) in the paper looks a bit unusual. “
> We did not see a performance difference using different resolutions (we tried 32x24 and 64x64) so we used smaller images for computational speed. We have included our numbers in the appendix, section A. Our resolution was based on parameters used in previous work on a similar environment (Shah et. al 2018).

---

### Official Review · AnonReviewer1 · 2018-11-03
**Limited technical merit and significance**

**Rating:** 5
**Confidence:** 4

**Review:**

This paper applies IRL to the cases of multiple tasks/environments and multimodal input features involving natural language (text) and vision (images). It is interesting to see the better performance of their proposed approaches with language-conditioned rewards over language-conditioned policies. The paper is written well.

I view the technical contributions of this work to be at best incremental; it does not seem to address any significant technical challenge to be able to integrate the various known tools in their work. I am not able to learn as much as i would have liked from this paper.

Considering the use of deep learning that can handle highly complex images and text, the practical significance of this work can be considerably improved by grounding their work in real-world context and/or larger-scale environments/tasks, as opposed to simulated environments in this paper. See, for example,

M. Wulfmeier, D. Rao, D. Z. Wang, P. Ondruska, and I. Posner, Large-scale cost function learning for path planning using deep inverse reinforcement learning, The International Journal of Robotics Research, 2017.

The authors say that "The work of MacGlashan et al. (2015) requires an extensively hand-designed, symbolic reward function class, whereas we use generic, differentiable function approximators that can handle arbitrary observations, including raw images." What then is the implication on how their proposed IRL algorithm is designed differently? How would the algorithm of MacGlashan et al. (2015) empirically perform as compared to the authors' proposed approach?



Minor issues

Page 2: a comparison to in Section 6 to as an oracle?
Page 3: What is rho_0?
Page 7: In order compare against?
Page 7: and and indicator on?

---

> ### Author Response · Authors · 2018-11-18
> **Thank you for your comments**
>
> Thank you for the review. We have responded to the concerns raised below by adding a comparison to another recent work that learns from raw images, clarifying the complexity of our task, and clarifying the contributions. We would appreciate it if the reviewer could revisit their review and let us know if any other concerns remain.
>
> “What then is the implication on how their proposed IRL algorithm is designed differently? How would the algorithm of MacGlashan et al. (2015) empirically perform as compared to the authors' proposed approach?”
> We added a comparison to another work (Bahdanau et al. (2018)) in Table 1 in Section 6.2, and to a modified version of GAIL (Ho et. al 2016), to provide a point of comparison. To our knowledge, this approach is closest in terms of problem statement, in that it also operates on raw RGB images. We unfortunately cannot compare to MacGlashan et al., due to a difference in the basic problem assumptions. The work of MacGlashan et al. require predefined  notions of “colors”, “objects”, “rooms” as well as semantic relationships between entities such as “roomIsRed” or “objectInRoom” to be given to the algorithm. In contrast, our method operates directly from raw text and image observations, making it easier to apply in practice without domain-specific engineering. We believe MacGlashan et. al is incomparable as it operates on significantly stronger assumptions, but we hope the baselines we have included are sufficient.
>
> “I view the technical contributions of this work to be at best incremental; it does not seem to address any significant technical challenge to be able to integrate the various known tools in their work. I am not able to learn as much as i would have liked from this paper.”
> As noted by R3,  the main technical challenge we address is to show the effectiveness of transferring reward over transferring policies across environments. Most works in IRL train and test within the same environment and do not concern themselves with generalization across environments. We provide a conceptually simple approach to a well-studied and important problem (language-based instruction following) that is relatively hyperparameter stable (we have included an ablation study over architectures) and easy to train. To our knowledge no previous works have applied deep inverse reinforcement learning to instruction-following tasks.
>
> “Considering the use of deep learning... larger-scale environments/tasks, as opposed to simulated environments in this paper.”
>
> Pertaining to the scale of our dataset, it contains 1413 tasks, and we use 10 demonstrations per task (14130 demonstrations total). Each task on average contains ~1200 states (for a total of ~1.7m states). As a comparison “Vision-and-Language Navigation: Interpreting visually-grounded navigation instructions in real environments” Anderson et. al. uses  7,189 demonstrations. Wulfmeier et. al reports 25,000 demonstrations used, but we could not find the number of states in their tasks.
>
> SUNCG is a commonly used dataset because it contains relatively realistic scenes. It is not a trivial or toy simulated task, but is commonly used for 3D visual navigation research in complex scenes (for example, FollowNet, Shah et. al 2018,  MINOS, Savva et. al. 2017). We believe that our tasks based on SUNCG are more realistic than many prior works on instruction following.

---

### Meta-Review · Area_Chair1 · 2018-12-14

**Confidence:** 4
**Recommendation:** Accept (Poster)

**Metareview:**

This paper generated a lot of discussion (not all of it visible to the authors or the public).

R1 initially requested reasonable comparisons, but after the authors provided a response (and new results), R1 continued to recommend rejecting the paper simply because they personally did not find the manuscript insightful. Despite several requests for clarification, we could not converge on a specific problem with the manuscript. Ungrounded gut feelings are not grounds for rejection.

After an extensive discussion, R2 and R3 both recommend accepting the paper and the AC agrees. Paper makes interesting contributions and will be a welcome addition to the literature.